# Tracking down the White Plague. Chapter three: Revision of endocranial abnormally pronounced digital impressions as paleopathological diagnostic criteria for tuberculous meningitis

Olga Spekker[1]*, David R. Hunt[2], William Berthon[1,3], László Paja[1], Erika Molnár[1], György Pálfi[1], Michael Schultz[4]

1 Department of Biological Anthropology, University of Szeged, Szeged, Hungary, 2 Department of Anthropology, National Museum of Natural History, Smithsonian Institution, Washington, District of Columbia, United States of America, 3 Chaire d'Anthropologie Biologique Paul Broca, École Pratique des Hautes Études (EPHE), PSL University, Paris, France, 4 Institut für Anatomie und Embryologie, Zentrum Anatomie, Universitätsmedizin Göttingen, Göttingen, Germany

☉ These authors contributed equally to this work.
* olga.spekker@gmail.com

## Abstract

Abnormally pronounced digital impressions (APDIs) on the endocranial surface develop secondary to a prolonged rise in the intracranial pressure. This can result from a number of pathological conditions, including hydrocephalus due to tuberculous meningitis (TBM). APDIs have been described with relation to TBM not only in the modern medical literature but also in several paleopathological studies. However, APDIs are not pathognomonic for TBM and their diagnostic value for identifying TBM in past human populations has not been evaluated in identified pre-antibiotic era skeletons. To assess the diagnostic value of APDIs for the first time, a macroscopic investigation was performed on skeletons from the Terry Collection (Smithsonian Institution, Washington, DC, USA). Our material consisted of 234 skeletons with tuberculosis (TB) as the cause of death (TB group) and 193 skeletons with non-tuberculous (NTB) causes of death (NTB group). The macroscopic examination focused on the stage of the prominence and frequency of APDIs in the TB group and NTB group. To determine the significance of difference (if any) in the frequency of APDIs between the two groups, $\chi^2$ testing of our data was conducted. We found that APDIs were twice as common in the TB group than in the NTB group. The $\chi^2$ comparison of the frequencies of APDIs revealed a statistically significant difference between the two groups. In addition, APDIs with more pronounced stages were recorded more frequently in the TB group. Our results indicate that APDIs can be considered as diagnostic criteria for TBM in the paleopathological practice. With suitable circumspection, their utilization provides paleopathologists with a stronger basis for identifying TB and consequently, with a more sensitive means of assessing TB frequency in past human populations.

**Data Availability Statement:** All relevant data are within the manuscript and its Supporting Information files.

**Funding:** This work was funded by the Hungarian State Eötvös Fellowship 2016 (77466) of the Tempus Public Foundation, the NTP-NFTÖ-16 (1116) of the Hungarian Ministry of Human Capacities & Human Capacities Grant Management Office, and the University of Szeged Open Access Fund (4932) to OS. The National Research, Development and Innovation Office (Hungary) (K 125561) provided funding for GP. The funders had no role in study design, data collection and analysis, decision to publish, or preparation of the manuscript.

**Competing interests:** The authors have declared that no competing interests exist.

## Introduction

Tuberculosis (TB) or the "White Plague" is one of the oldest known infectious diseases that has been affecting humanity for millennia [1–3]. Traditionally, signs of different forms of osteoarticular TB (e.g., spinal TB and/or TB arthritis of the large, weight-bearing joints) have been considered to establish the paleopathological diagnosis of the disease [4–9]. Nevertheless, skeletal TB can be identified in only about 3–5% of all the TB cases in past human populations; thus, it is difficult to estimate the true TB frequency in human osteoarchaeological series if we consider only the aforementioned diagnostic criteria [4,5,9,10]. What would be more helpful would be to identify further macromorphological diagnostic criteria for TB to provide a more relevant disease frequency in past human populations. To contribute to the establishment of a more reliable and accurate paleopathological diagnosis of TB, numerous studies [e.g., 4–24] were performed on osteoarchaeological series and documented skeletal collections since the 1980s. As a result of these investigations, a positive association between TB and several pathological bony changes, including endocranial abnormally pronounced digital impressions (APDIs), has been recognized.

Digital impressions (DIs) on the endocranial surface are shallow depressions (resembling finger imprints) that correspond to the cerebral gyri [25–30]. DIs are incompletely separated from each other by bony ridges of different sizes (i.e., the cerebral juga) that match in their position to the cerebral sulci [27,29,30]. DIs originate under physiological conditions from brain growth and cortical folding (i.e., gyrification) in childhood and early adolescence [25–31]. The formation of DIs is a result of temporary, circumscribed bone resorption: the localized pressure exerted by the brain and its pulsating blood vessels on the underlying endocranial surface induces pressure atrophy of the bone [26–34]. Although very pronounced DIs–generally confined to the skull base and to the lower two-thirds of the skull vault–may be normal in subadults (particularly during periods of rapid brain growth), the prominence of DIs decreases during adolescence [28–31,33–37]. In adulthood, the presence of APDIs over the upper portion of the skull indicates a prolonged rise in the intracranial pressure: according to estimates, the formation of APDIs secondary to elevated intracranial pressure (eICP) requires at least ten weeks [29,30,32–36]. Besides many other pathological conditions that can affect the endocranium due to space-consuming processes [e.g., 38–40]–such as central nervous system (CNS) infections other than tuberculous meningitis (TBM) [e.g., 41–43], traumata [e.g., 34,44,45], brain tumors [e.g., 34,38–40,46] or hemorrhages [e.g., 34,38,43,44,46])–, TBM can also lead to the development of APDIs on the endocranial surface by resulting in eICP [12,14,15,47–49]. Although APDIs were described in relation to TBM in the paleopathological literature [12,14,15], they are not pathognomonic traits of the disease and still represent a relatively poorly researched area of paleopathology today: previous research papers [e.g., 12–15,19] did not assess the diagnostic value of APDIs and/or did not include statistical analyses.

Scientific studies from the pre-antibiotic era describe, by clinical pictures and drawings, morphological features of a disease that have changed substantially to the form today. These lesions have often been considered as pathognomonic. Furthermore, the descriptions of morphological changes, especially in the skeletal system, from these earlier handbooks are particularly rich in detail. So, the pre-antibiotic era literature is extremely helpful for the interpretation of vestiges of diseases for paleopathologists and forensic anthropologists. Besides the meticulous descriptions from the end of the 19[th] century and the first half of the 20[th] century, the comprehensive macroscopic investigation of skeletons deriving from identified pre-antibiotic era collections, such as the Hamann–Todd Human Osteological Collection, the Robert J. Terry Anatomical Skeletal Collection, the Stanford University Medical School Collection, and the Coimbra Identified Skeletal Collection, can serve as a unique and

important basis for determining the diagnostic value of probable TB-related bony changes, including APDIs, in the paleopathological identification of TB [10,50]. Although living patients can only be surveyed with medical imaging techniques, skeletons from documented anatomical collections can directly be evaluated with macromorphological methods [5,6]. Furthermore, in identified pre-antibiotic era skeletons, the manifestation of TB and consequently, the appearance of likely TB-related bony changes can be similar to those of observable in human osteoarchaeological materials. This is different for living patients with TB, since their received antibiotic therapy can alter the manifestation of the disease [4–7]. From the 1980s, several identified skeletal collections have been used to define and refine macromorphological diagnostic criteria for TB that can be applied in the paleopathological practice [e.g., 4–10,18–20,22,51–54]. Nonetheless, APDIs were beyond the scope of the above-mentioned research projects. Therefore, a comprehensive macromorphological study regarding APDIs on pre-antibiotic era skeletons of known cause of death from documented collections is still needed to assess the diagnostic value of APDIs for identifying TBM in human osteoarchaeological series.

The main aim of our study is to expand the knowledge and understanding on the development of APDIs, as well as to improve their paleopathological interpretation along with strengthening their diagnostic value in the identification of TBM in past human populations. This is accomplished by 1) providing a solid background in the development of APDIs based on the medical literature from the pre-antibiotic era and today, and 2) presenting results of the detailed macroscopic investigation that, for the first time, focused on the macromorphological characteristics (i.e., the stage of the prominence of lesions) and the frequency of APDIs in selected late adolescent and adult skeletons of known cause of death from the Terry Collection.

The objectives of our paper are:

1. To macroscopically evaluate the selected skeletons from the Terry Collection for the presence of APDIs;

2. To compare the frequencies of APDIs between individuals recorded to have died of TB versus those identified to have died of causes other than TB;

3. To macromorphologically characterize APDIs regarding the stage of the prominence of lesions on the affected cranial bone(s); and

4. To evaluate the diagnostic value of APDIs.

## Materials and methods

### Materials

The Robert J. Terry Anatomical Skeletal Collection consists of 1,728 human skeletons (1,011 males and 717 females), mostly coming from the pre-antibiotic era, that are housed at the Smithsonian Institution Museum Support Center (Suitland, MD, USA) as part of the collections of the Department of Anthropology, National Museum of Natural History (Smithsonian Institution, Washington, DC, USA).The skeletons were accumulated first by Robert J. Terry (professor of anatomy and head of the Anatomy Department at Washington University Medical School in St. Louis, MO, USA) from the second decade of the 20th century until his retirement in 1941. Mildred Trotter, who succeeded Terry as anatomy professor, continued Terry's work between 1941 and 1967. In the Terry Collection, individuals were born between 1828 and 1943 and died between 1905 and 1966, with age at death ranging from 14 to 102 years. Owing to Terry's well-established uniform protocol for the collecting, cataloguing, maceration, and storage of bone remains, almost all of the skeletons in the Terry Collection are complete and well-preserved. Moreover, a series of documentary forms (e.g., morgue record, dental

chart, anthropometric and anthroposcopic data form, and bone inventory list) providing various biographical information (e.g., name, sex, age at death, "race", occupation, and cause of death) and basic anthropological data is available for each individual. On account of the above, the Terry Collection presents an invaluable resource for anthropological and medical research, including paleopathological studies that attempt to define or refine diagnostic criteria for the identification of specific infectious diseases (such as TB) in past human populations [50].

As part of a comprehensive research project [55], all individuals (N = 302) recorded to have died of different types of TB (e.g., pulmonary TB, miliary TB, peritoneal TB, and skeletal TB) from the Terry Collection were evaluated for the macroscopic characteristics, frequencies, and co-occurrences of different types of pathological skeletal lesions likely associated with TB. In addition, 302 randomly selected individuals from the remaining skeletons of the Terry Collection, recorded to have died of causes other than TB (e.g., other infectious diseases, cardiovascular problems, cancer, and external causes, such as car accident, suicide or homicide), were also assessed for the same aspects. It is possible that some of these individuals were also afflicted with TB but the morgue record and/or the death certificate did not indicate the disease as primary, secondary or tertiary cause of death. Although a total of 604 individuals were surveyed in the Terry Collection in the current research project, not all of them were suitable to the examination considering APDIs. The skullcap was missing in two cases, the skull was not sectioned in a further 173 cases, and age at death was uncertain in two additional cases; therefore, the aforementioned 177 skeletons were excluded from the macromorphological and statistical analyses regarding APDIs. The remaining sample consisted of seven late adolescent (16–19 years old; three males and four females) and 420 adult ($\geq$20 years old; 272 males and 148 females) individuals with skulls sectioned in the transverse plane and occasionally also in the mid-sagittal plane. These individuals were divided into two main groups on the basis of their causes of death. The TB group was composed of 234 individuals identified to have died of TB (169 males and 65 females), with age at death ranging from 16 to 81 years (S1 Table) [23,24]; whereas the control or non-TB (NTB) group consisted of 193 individuals recorded to have died of causes other than TB (106 males and 87 females), with age at death ranging from 20 to 90 years (S2 Table) [23,24].

## Ethics statement

Individual numbers: T4R, T12R, T13R, T19R, T23R, T25, T25R, T30R, T31R, T35R, T39, T44R, T46R, T47R, T54, T58R, T62RR, T64R, T69, T76R, T79R, T84, T87R, T89R, T90, T91R, T95, T95R, T103R, T104RR, T105R, T112R, T114, T124R, T127R, T128, T129, T130, T132R, T134, T135R, T138, T139, T140RR, T141R, T142R, T145R, T146R, T149R, T158R, T167, T177R, T178R, T179R, T182, T194, T197R, T199, T200, T204, T205, T207, T218, T220, T221, T222, T227, T230, T231, T232R, T235, T237, T243R, T247R, T248R, T249R, T250, T251, T254, T255, T259, T264, T265, T267, T268, T269, T270, T272, T279, T280, T282, T283R, T284, T285, T293R, T296R, T298, T304, T306, T306R, T309, T314, T317, T318, T328R, T329, T338, T339R, T341, T344R, T347, T348R, T353, T358R, T382R, T385, T386R, T393RR, T397, T400, T402, T403, T410R, T422, T423, T424, T426R, T432, T437R, T438, T444, T445, T447, T452, T453, T458, T463, T465, T466, T468, T470, T477, T483, T490, T496, T497, T504, T506, T512, T513RR, T522, T523, T527, T528, T534, T536, T537, T541, T545, T549, T552, T555, T562, T565, T566, T568, T571, T572, T573, T575, T582, T583, T585, T586, T592, T595, T597, T602, T608, T617R, T620, T621R, T626R, T627R, T629, T636, T657R, T664, T669R, T679, T680, T686, T694, T702R, T726, T727, T728R, T739, T752, T757, T759, T761, T771, T776, T786, T789, T795, T799, T809R, T820R, T822, T823, T828, T833R, T834R, T844, T846, T863, T876, T891, T892, T895, T896RR, T897, T902, T903R, T907, T914, T915, T919, T930R, T932,

T933R, T934, T936, T938, T941, T946, T948, T950, T952, T955, T957, T957R, T964, T968, T973, T975, T987, T1002, T1005, T1013, T1018, T1020, T1023, T1027, T1029R, T1030, T1031, T1033, T1034, T1036, T1043, T1045, T1046, T1047, T1048, T1050, T1057, T1058, T1060, T1066R, T1070, T1071R, T1072, T1076, T1086, T1093, T1095, T1096R, T1098, T1100RR, T1102R, T1105, T1106, T1107, T1113, T1122, T1124R, T1129, T1130R, T1132, T1133RR, T1134R, T1137R, T1138R, T1140, T1147R, T1156, T1157, T1159, T1163, T1165, T1169, T1173, T1182R, T1183, T1185, T1186, T1187, T1190, T1192, T1204R, T1205, T1210, T1215, T1219, T1222, T1224, T1226, T1228, T1229, T1230, T1232, T1236, T1243R, T1247, T1249R, T1252, T1255, T1263R, T1264, T1266R, T1267, T1271, T1275, T1277, T1278, T1282, T1285, T1287, T1291, T1299R, T1300, T1304R, T1309, T1310, T1313, T1315, T1318, T1319, T1322, T1331, T1337RR, T1342, T1343, T1346, T1347R, T1352, T1353R, T1359, T1362, T1367, T1368, T1369, T1375R, T1376, T1377, T1378, T1379, T1387, T1388, T1397, T1398, T1401, T1405, T1406, T1407, T1411R, T1416, T1417R, T1419, T1422R, T1428R, T1434R, T1435, T1439R, T1444, T1451, T1453R, T1455, T1458, T1467, T1476, T1495, T1502R, T1503, T1505R, T1507, T1514, T1519, T1521, T1531, T1533, T1534, T1536, T1539, T1543, T1544, T1549, T1551, T1552, T1553, T1554, T1555, T1562, T1567, T1568, T1572, T1576, T1581, T1592, T1599, T1604, T1614, T1627, and T1629.

All human skeletal remains assessed in the described study are housed at the Smithsonian Institution Museum Support Center (Suitland, MD, USA) as part of the collections of the Department of Anthropology, National Museum of Natural History (Smithsonian Institution, Washington, DC, USA). Access to these human skeletal remains is granted by the Department of Anthropology.

No permits were required for the described study, complying with all relevant regulations.

## Methods

All of the 427 selected skeletons from the Terry Collection were macromorphologically evaluated for the presence of APDIs. During the macroscopic investigation, the study personnel had no information on the cause of death of the examined individuals so as to reduce the risk of bias. A special data collection sheet was developed for the current research project, on which detailed written and pictorial descriptions were made for each individual. Two evaluation rounds were performed on the endocranial surface of the 427 selected skeletons. In the first evaluation round, the examined individuals were divided into two categories based on the presence of APDIs (i.e., present or not present) (Table 1), and three cases with APDIs representing the most "typical" of the different stages of the lesion prominence were chosen as reference cases. The three reference cases are the following:

**Table 1. Classification criteria for the presence of APDIs, considering the different prominence stages.**

| Presence and prominence stages of APDIs | | Classification category | Detailed description |
|---|---|---|---|
| Not present | | 0 | No APDIs are present on the inner skull surface. |
| Present | Very slight prominence stage | 1 | APDIs are present on the inner surface of the skullcap–only on the squamous part of the frontal bone (Fig 1A). |
| | Slight prominence stage | 2 | APDIs are present on the inner surface of the skullcap–on the squamous part of the frontal bone and on the two parietal bones (Fig 1B). |
| | Pronounced prominence stage | 3 | APDIs are present not only on the skullcap but also on the skull base–besides on the squamous part of the frontal bone and on the two parietal bones, on the squamous part of the two temporal bones and/or the occipital bone (Fig 1C and 1D). |

1. Terry No. 30R (26-year-old, male, died of TBM)–very slight stage: shallow DIs predominantly over the anterior portion of the inner skull surface of the skullcap (i.e., on the squamous part of the frontal bone) (Fig 1A);

2. Terry No. 382R (26-year-old, male, died of pulmonary TB)–slight stage: deeper DIs particularly over the anterior and middle portions of the endocranial surface of the skullcap (i.e., on the squamous part of the frontal bone and on the two parietal bones) (Fig 1B); and

3. Terry No. 1033 (26-year-old, male, died of pulmonary TB)–pronounced stage: deep DIs all over the inner skull surface (i.e., not only on the skullcap but also on the skull base) (Fig 1C and 1D).

In the second evaluation round, individuals showing APDIs on the endocranial surface were compared to the three reference cases and further classified based on the prominence of APDIs (i.e., very slight, slight or pronounced) (Table 1). During the macromorphological assessment of the 427 selected skeletons, the co-occurrence of APDIs with other pathological bony changes likely related to different forms of TB (e.g., pulmonary TB/TB pleurisy [e.g., 4–8,18,19,51–54], skeletal TB [e.g., 9,10,19,20,56–59], and TBM [e.g., 12–15,19,21,23,24]) was also recorded (Table 2).

Prior to the statistical data analysis, all information collected during the detailed macroscopic investigation was entered into a Microsoft Office Excel spreadsheet. Firstly, absolute and percentage frequencies of APDIs were calculated in both the TB group and NTB group, considering the stage of the prominence of APDIs. Secondly, to determine the significance of differences (if any) in frequencies of APDIs between the two groups, $\chi^2$ testing of the data was conducted, using the MedCalc statistical software package. Finally, to establish the diagnostic efficacy of APDIs, sensitivity and specificity estimate values were generated, using the MedCalc statistical software package.

## Results

From a total of 427 skeletons assessed in the Terry Collection, 216 (50.59%) displayed APDIs on the endocranial surface (Table 3): 154 (65.81%) of 234 individuals identified to have died of TB (S1 Table) and 62 (32.12%) of 193 individuals recorded to have died of causes other than TB (S2 Table). The $\chi^2$ comparison revealed a statistically extremely significant difference in the frequencies of APDIs between the TB group and NTB group ($\chi^2$ = 46.680, df = 1, P<0.0001) (Table 3). The generated sensitivity and specificity estimate values for APDIs were 65.81% (95% CI: 59.35% to 71.87%) and 67.88% (95% CI: 60.79% to 74.40%), respectively (Table 4).

Of the 154 individuals with APDIs in the TB group, 124 died of pulmonary TB (S1 Table). Nine additional individuals were registered to have died of other types of tuberculosis, such as skeletal TB (four cases), peritoneal TB (two cases), TBM (two cases), and miliary TB (one case); whereas, in the remaining 21 cases, the type of tuberculosis as the cause of death was not specified on the morgue record and/or death certificate (S1 Table). In the NTB group, the most frequently recorded NTB causes of death were cardiovascular problems, followed by respiratory diseases, infectious diseases other than TB, and different types of cancer among individuals revealing APDIs on the inner skull surface (S2 Table).

Regarding the stage of the prominence of APDIs detected, from the 216 skulls affected, 148 (68.52%), 51 (23.61%), and 17 (7.87%) represented the very slight (Figs 1A and 2A), slight (Figs 1B and 2B), and pronounced (Figs 1C and 1D and 2C) stages, respectively (Fig 3 and 3 and S1 and S2 Tables). Although the very slight stage of the prominence of APDIs was more common in individuals with NTB causes of death (Fig 3B) and the more pronounced (i.e.,

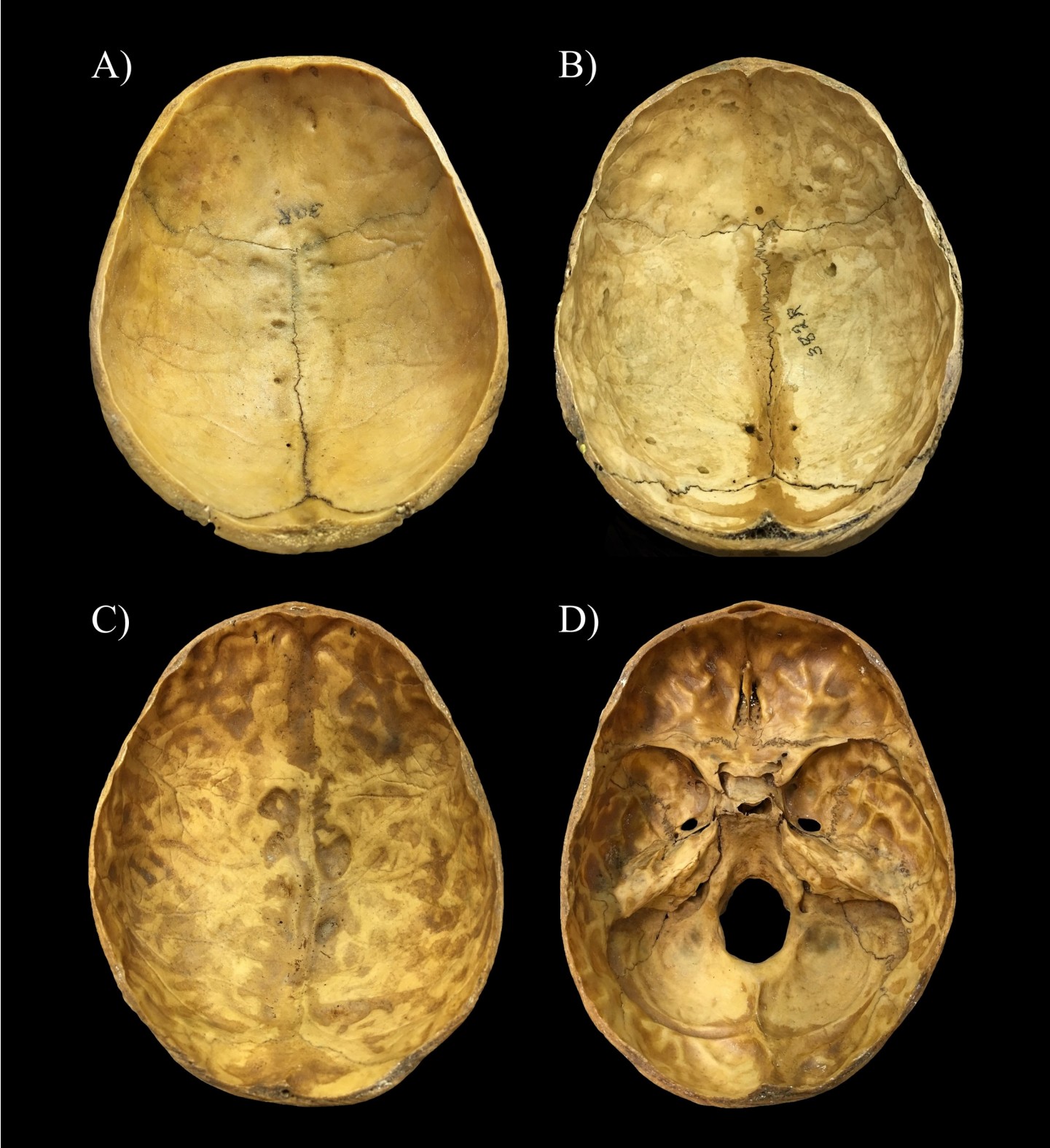

**Fig 1. Reference cases selected for the classification of individuals exhibiting APDIs on the endocranial surface of the skull.** A) Very slight stage represented by shallow DIs predominantly localized over the anterior portion of the inner skull surface (Terry No. 30R: 26-year-old, male, died of TBM–skullcap); B) Slight stage represented by deeper DIs particularly situated over the anterior and middle portions of the endocranial surface (Terry No. 382R: 26-year-old, male, died of pulmonary

TB–skullcap); C) Pronounced stage represented by deep DIs localized all over the inner skull surface (Terry No. 1033: 26-year-old, male, died of pulmonary TB–skullcap); and D) Pronounced stage represented by deep DIs situated all over the endocranial surface (Terry No. 1033: 26-year-old, male, died of pulmonary TB–skull base).

slight and pronounced) stages of the prominence of APDIs occurred more frequently in individuals with TB as the cause of death (Fig 3A), statistically significant difference between the two groups was found only in the frequencies of the very slight stage of the prominence of APDIs (very slight stage: $\chi^2 = 8.530$, df = 1, P = 0.0035; slight stage: $\chi^2 = 3.312$, df = 1, P = 0.0688; and pronounced stage: $\chi^2 = 3.564$, df = 1, P = 0.0591) (Table 3).

In 142 of the 154 individuals (92.21%) with APDIs in the TB group (Fig 4A and Table 5), APDIs simultaneously occurred with other likely TBM-associated endocranial alterations (S3 Table) and/or probable TB-related non-endocranial bony changes (S5 Table):

- Abnormal blood vessel impressions: 36 cases;

- Periosteal appositions: 50 cases;

- Granular impressions: 49 cases;

- Periosteal new bone formations on the visceral surface of ribs: 95 cases;

- Vertebral hypervascularization: 87 cases;

- Vertebral lytic lesions: 18 cases;

- Signs of extra-spinal osteomyelitis: 10 cases;

- Signs of extra-spinal arthritis: 10 cases;

- Signs of hypertrophic pulmonary osteopathy: 16 cases; and

- Signs of cold abscesses: 21 cases.

**Table 2. List of likely TB-related pathological bony changes other than APDIs that were considered during the evaluation of the 604 selected skeletons from the Terry Collection.**

| Form of TB | | Pathological skeletal lesion type |
|---|---|---|
| **Pulmonary TB/TB pleurisy** | | Periosteal new bone formations and/or erosive lesions on the ribs (predominantly on the visceral costal surfaces) [e.g., 4–8,19] |
| | | Signs of diffuse, symmetrical periostitis on the diaphysis of short and/or long tubular bones (i.e., hypertrophic pulmonary osteopathy) [e.g., 18,19,50–52] |
| **Skeletal TB** | **Spinal TB** | Osteolytic and/or erosive changes on the vertebral bodies and/or posterior elements [e.g., 9,19,56–57,59] |
| | | Destruction, collapse, and/or fusion of the vertebral bodies [e.g., 9,10,19,56,57,59] |
| | | Signs of hypervascularization on the anterior and/or lateral aspects of the vertebral bodies [e.g., 9,10,19,20,58] |
| | | Cortical remodeling and/or reactive new bone formations on the vertebral surfaces [e.g., 9,19,20,57,59] |
| | | Cortical remodeling and/or reactive new bone formations on the sacrum, hip bones, and/or femora (i.e., signs of an overlying cold abscess) [e.g., 9,57,59] |
| | | Destruction, subluxation, and/or dislocation of the intervertebral joints [e.g., 9,57,59] |
| | **Extra-spinal TB osteomyelitis** | Osteolytic and/or erosive changes on the extra-spinal bones [e.g., 9,19,20,57,59] |
| | | Cortical remodeling and/or reactive new bone formations on the extra-spinal bones [e.g., 9,19,20,57,59] |
| | **Extra-spinal TB arthritis** | Destruction, subluxation, and/or dislocation of the extra-spinal joints [e.g., 19,57,59] |
| **TB meningitis** | | Granular impressions on the inner skull surface [e.g., 12–15,19,21,23] |
| | | Abnormal blood vessel impressions on the inner skull surface [e.g., 12–15,19,24] |
| | | Periosteal appositions on the inner skull surface [e.g., 12–15,19,24] |

**Table 3. Summary of the statistical results, considering the different stages of the prominence of APDIs.**

| Presence of APDIs by prominence stage | | TB group ($N_{TB}$ = 234) | | NTB group ($N_{NTB}$ = 193) | | $\chi^2$-probe | | |
|---|---|---|---|---|---|---|---|---|
| | | n | % | n | % | $\chi^2$ | df | P |
| Very slight stage | Present | 96 | 41.03 | 52 | 26.94 | 8.530 | 1 | 0.0035 |
| | Not present | 138 | 58.97 | 141 | 73.06 | | | |
| Slight stage | Present | 42 | 17.95 | 9 | 4.66 | 3.312 | 1 | 0.0688 |
| | Not present | 192 | 82.05 | 184 | 95.34 | | | |
| Pronounced stage | Present | 16 | 6.84 | 1 | 0.52 | 3.564 | 1 | 0.0591 |
| | Not present | 218 | 93.16 | 192 | 99.48 | | | |
| Any of the three stages | Present | 154 | 65.81 | 62 | 32.12 | 46.680 | 1 | <0.0001 |
| | Not present | 80 | 34.19 | 131 | 67.88 | | | |

In 28 out of the 62 individuals (45.16%) with APDIs in the NTB group, likely TBM-associated endocranial alterations other than APDs (S4 Table) and/or probable TB-related non-endocranial bony changes (S6 Table) were concomitantly recorded (Fig 4B and Table 5):

- Abnormal blood vessel impressions: 3 cases;

- Periosteal appositions: 7 cases;

- Granular impressions: 4 cases;

- Periosteal new bone formations on the visceral surface of ribs: 4 cases;

- Vertebral hypervascularization: 12 cases;

- Vertebral lytic lesions: 3 cases;

- Signs of extra-spinal osteomyelitis: 3 cases;

- Signs of hypertrophic pulmonary osteopathy: 4 cases; and

- Signs of cold abscesses: 2 cases.

## Discussion and conclusions

In the paleopathological literature [e.g., 12,14,15], endocranial APDIs developing secondary to a prolonged rise in the intracranial pressure [28,30,32,35,36] have been described in relation to TBM. However, it should not be overlooked that APDIs can also occur clinically in other meningeal disorders [e.g., 38,60–63], as well as in association with several brain diseases [e.g., 38].

**Table 4. Summary of the statistical results, considering the diagnostic sensitivity and specificity of APDIs.**

| APDIs | Recorded to have died of TB ($N_{TB}$ = 234) | Recorded to have died of NTB causes ($N_{NTB}$ = 193) |
|---|---|---|
| Present | 154 (i.e., true positives or TP) | 62 (i.e., false positives) |
| Not present | 80 (i.e., false negatives) | 131 (i.e., true negatives or TN) |
| Sensitivity (TP/$N_{TB}$) | 154/234 = 0.6581 (65.81%) | |
| 1 –Sensitivity | 0.3419 (34.19%) | |
| Specificity (TN/$N_{NTB}$) | 131/193 = 0.6788 (67.88%) | |
| 1—Specificity | 0.3212 (32.12%) | |
| Sensitivity + specificity | 1.3369 | |

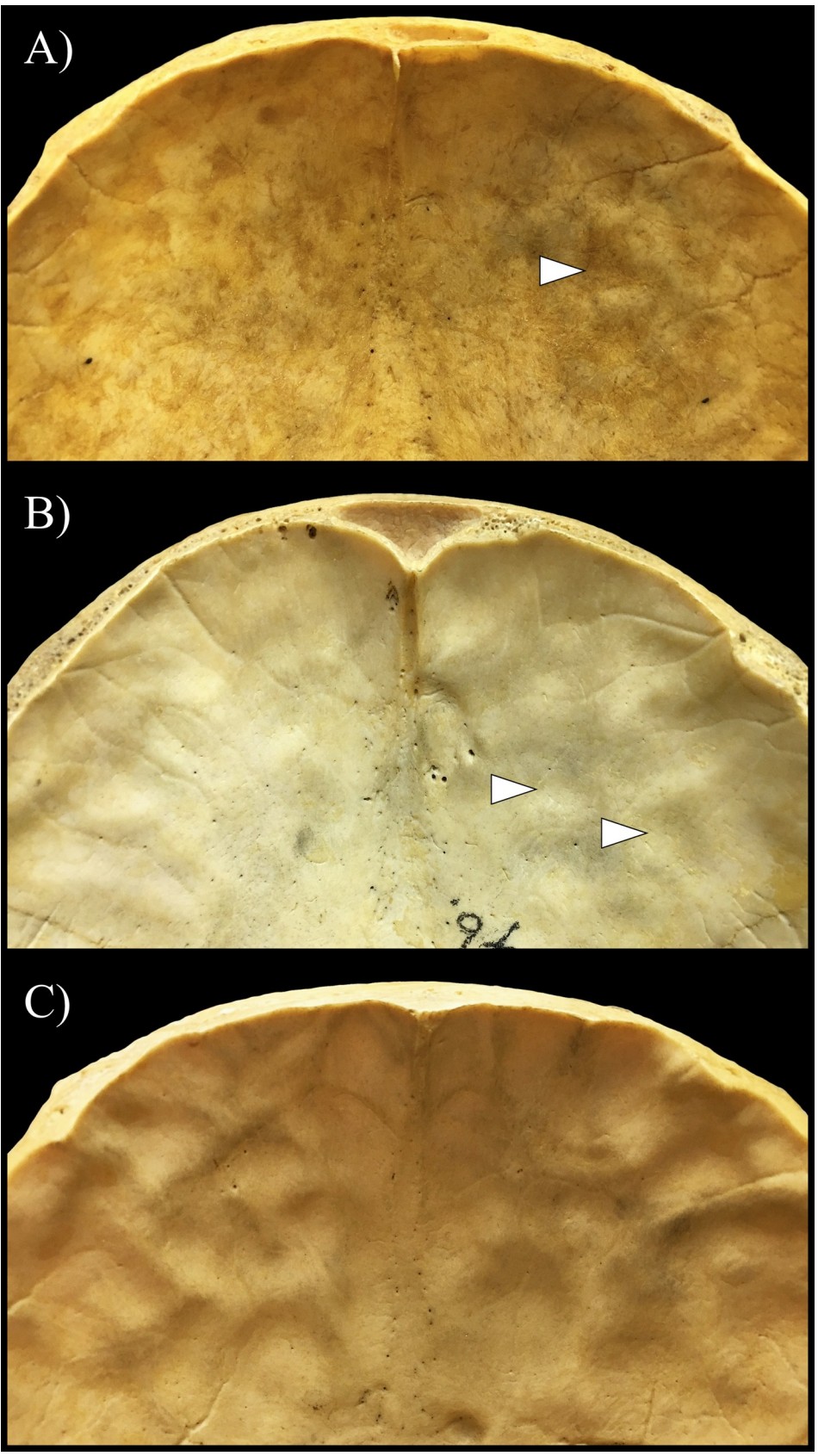

**Fig 2. Different stages of the prominence of APDIs (white arrows) on the inner surface of the squamous part of the frontal bone.** A) Very slight stage represented by shallow DIs on the endocranial surface of the frontal bone of Terry No. 1036 (38-year-old, male, died of pulmonary TB); B) Slight stage represented by deeper DIs on the inner skull surface of the frontal bone of Terry No. 265 (32-year-old, male, died of TB); and C) Pronounced stage represented by deep DIs on the endocranial surface of the frontal bone of Terry No. 251 (34-year-old, male, died of pulmonary TB).

TBM is characterized by diffuse granulomatous inflammation of the leptomeninges (i.e., the pia and arachnoid mater encephali), with strong predilection for the basal areas of the brain [64–71]. In later stages of the disease, the outermost meningeal layer (i.e., the dura mater encephali) can also be affected [72,73]. Besides the small tubercles primarily formed in the leptomeninges and later also in the dura mater encephali, enhancing basal meningeal exudate, progressive hydrocephalus, and vasculitis of the blood vessels adjacent to or traversing the exudate are also characteristic pathological features of TBM [43,68,70–78].

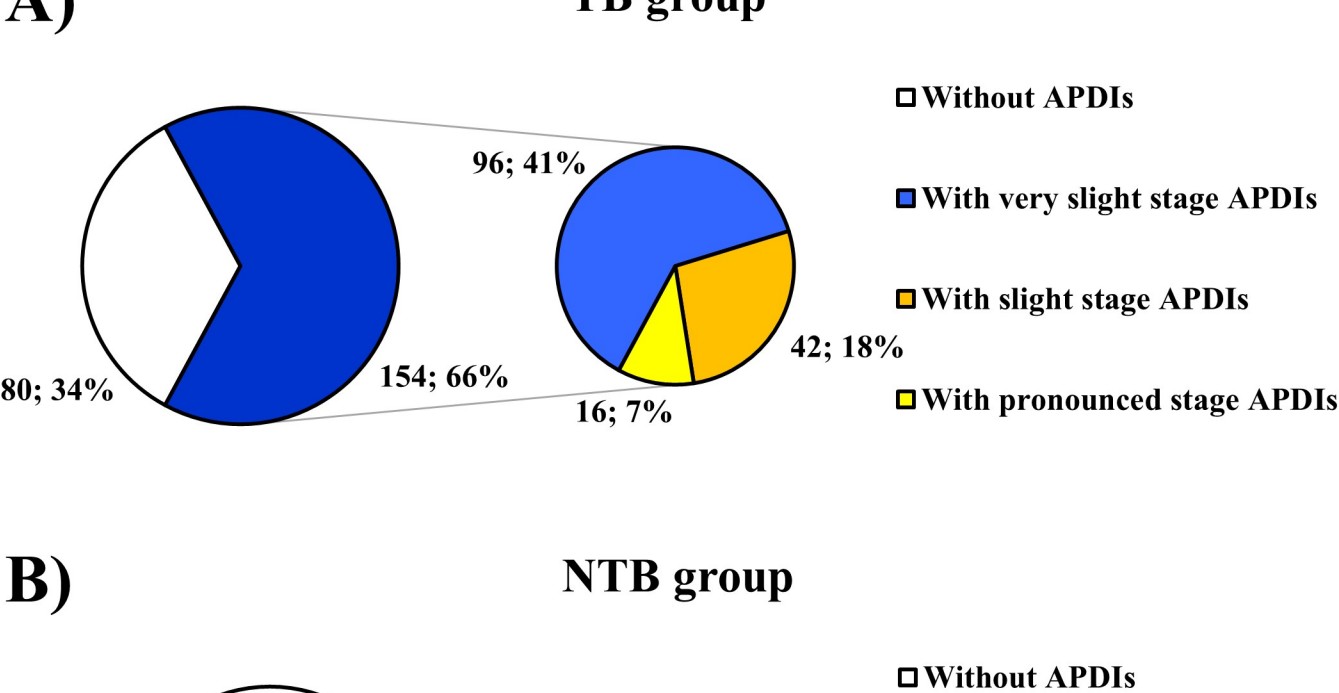

**Fig 3. Distribution of the selected individuals from the Terry Collection by the presence of APDIs (considering the different stages of the prominence of lesions).** A) Distribution of individuals who died of TB; and B) Distribution of individuals who died of NTB causes. (In both the TB group (A) and NTB group (B), the larger pie chart shows the distribution of all individuals by the presence of APDIs (i.e., not present (white) or present (dark blue)). Whereas, the smaller pie chart shows the distribution of individuals displaying APDIs by the prominence stage of the presented APDIs (i.e., very slight (light blue), slight (orange), and pronounced (yellow)). The percentage values were calculated with respect to the total number of individuals in both the TB group ($N_{TB}$ = 234) and NTB group ($N_{NTB}$ = 193)).

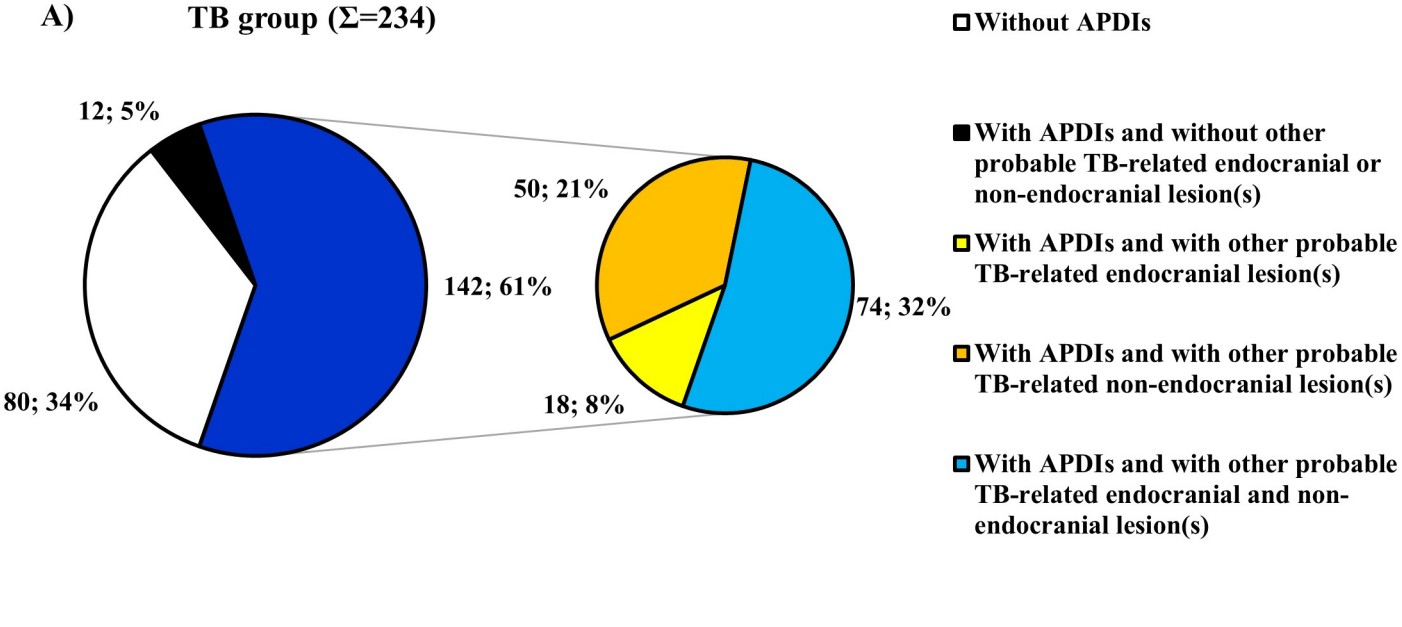

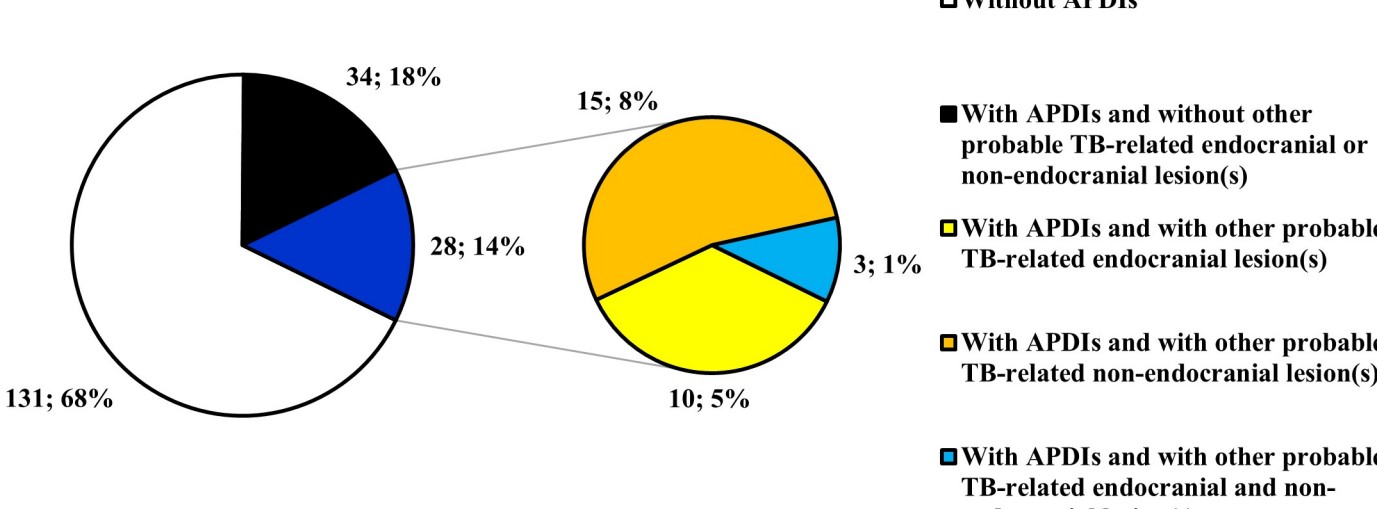

**Fig 4. Distribution of the selected individuals from the Terry Collection by the presence of APDIs (considering the co-occurrence of APDIs with other probable TBM-related endocranial lesion types and/or with possible TB-associated non-endocranial lesion types).** A) Distribution of individuals who died of TB; and B) Distribution of individuals who died of NTB causes. (In both the TB group (A) and NTB group (B), the larger pie chart shows the distribution of all individuals by the presence of APDIs (i.e., not present (white), present but alone (black) or present in association with other likely TB-related lesion(s) (dark blue)). The smaller pie chart shows the distribution of individuals displaying APDIs in association with other probable TB-related lesion(s) by the type of the lesion(s) simultaneously occurring with APDIs (i.e., besides APDIs, only other probable TBM-related endocranial lesion(s) are present (yellow), besides APDIs, only possible TB-associated non-endocranial lesion(s) are present (orange) or besides APDIs, both other probable TBM-related endocranial lesion(s) and possible TB-associated non-endocranial lesion(s) are present (light blue)). The percentage values were calculated with respect to the total number of individuals in both the TB group ($N_{TB}$ = 234) and NTB group ($N_{NTB}$ = 193)).

In the initial stage of TBM, the granulomatous inflammatory reaction results in the formation of a thick, gelatinous exudate between the two layers of the leptomeninges [64,66,67,69,72,75,79,80]. The inflammatory exudate is primarily located along the

**Table 5. Summary of the statistical results, considering the co-occurrence of APDIs with other likely TB-related lesion types.**

| Categories | Individuals with APDIs in the TB group (N = 154) | | Individuals with APDIs in the NTB group (N = 62) | |
|---|---|---|---|---|
| | n | % | n | % |
| Besides APDIs, no other likely TB-related lesions are present | 12 | 7.79 | 34 | 54.84 |
| Besides APDIs, only other probable TBM-related endocranial lesions are present | 18 | 11.69 | 10 | 16.13 |
| Besides APDIs, only possible TB-associated non-endocranial lesions are present | 50 | 32.47 | 15 | 24.19 |
| Besides APDIs, both other probable TBM-related endocranial lesions and possible TB-associated non-endocranial lesions are present | 74 | 48.05 | 3 | 4.84 |

inferomedial surface of the frontal lobes, the anteromedial surface of the temporal lobes, the floor of the third ventricle, and the superior aspect of the cerebellum [64,75,77,79,81]. From here, it can rapidly extend towards the basal cisterns (i.e., the interpeduncular and chiasmatic cisterns) [64,74,75,77,78,80–82]. As the disease progresses, the infection can gradually spread to other cisterns, such as the prepontine, ambient, and Sylvian cisterns, and eventually, can reach the meninges covering the cerebral convexities, the ependymal surface of the ventricles, and the choroid plexuses [64,66,74,75,77,78,80–82].

The inflammatory exudate, partially or completely filling the subarachnoid space and/or the ventricular pathways, can lead to the development of persistent and progressive internal hydrocephalus (i.e., disturbance of the flow, reabsorption or production of the cerebrospinal fluid (CSF) that leads to an increase in volume occupied by the CSF in the CNS) [39,42,60,63,64,81–88]. Hydrocephalus is one of the most common complications of TBM that occurs in more than two-thirds of the cases [64,75,81,82,84–88]. In TBM, either the communicating or the non-communicating type of hydrocephalus can develop, with the former being more common [64,66,75,77,78,82,84,85,87]. Communicating TBM-related hydrocephalus usually occurs when the inflammatory exudate blocks the CSF flow within the subarachnoid space, consequently resulting in impaired CSF reabsorption [64,66,69,75,77,81,82,84,85,87]. In later stages of the disease, inflammation of the ependymal surface of the ventricles and of the choroid plexuses leads to CSF overproduction, also contributing to the progression of communicating TBM-related hydrocephalus [82,84,85]. The non-communicating type of TBM-related hydrocephalus develops when the inflammatory exudate obstructs the pathways connecting the ventricles (i.e., the interventricular foramina of Monro and the cerebral aqueduct of Sylvius) or the passages between the fourth ventricle and the subarachnoid space (i.e., the lateral apertures of Luschka and the median aperture of Magendie), consequently resulting in blockage of the CSF flow [64,66,69,74,75,79,81,82,84,87]. TB hydrocephalus is often associated with eICP [82,87]; and therefore, besides other pathological conditions (e.g., CNS infections other than TBM, brain tumors, and hemorrhages), TBM can result in the development of APDIs on the endocranial surface.

Since APDIs cannot be considered as pathognomonic features of TBM, their utilization as diagnostic criteria for the disease in the paleopathological practice can be questioned, especially in consideration that previous studies [e.g., 12,14,15,19] did not assess the diagnostic value of APDIs on identified pre-antibiotic era skeletons and/or did not include statistical data analysis. To fill the aforementioned research gap, we performed a macroscopic investigation on identified skeletons from the Terry Collection that focused on the macromorphological characteristics and frequency of APDIs and was completed by subsequent statistical analysis of data.

Of the 427 selected skeletons with sectioned skulls from the Terry Collection, about one-half exhibited APDIs on the inner skull surface and APDIs were recorded in both the TB

group and NTB group. Our findings indicate that APDIs do have a diagnostic value in the paleopathological identification of TBM in ancient human bone remains, since they were about twice more common in individuals with TB as the cause of death than in individuals with NTB causes of death. Furthermore, the $\chi^2$ comparison of the frequencies of APDIs revealed a statistically extremely significant difference between the TB group and NTB group (Table 3). This result suggests a positive association between APDIs and TBM. In addition to the above, in the vast majority of the 154 individuals with APDIs in the TB group, APDIs simultaneously occurred with other probable TBM-related endocranial alterations (Fig 4A and Tables 5 and S3) and/or possible TB-associated non-endocranial bony changes (Fig 4A and Tables 5 and S5). This observation further supports the tuberculous origin of APDIs observed in the TB group.

Our results fit with those of previous studies [e.g., 12–15,19] concerning the specificity of APDIs for TBM. APDIs were registered in about one-third of the skeletons composing the NTB group. This indicates that APDIs cannot be considered as specific vestiges of TBM. In about one-half of the 62 individuals with APDIs in the NTB group, probable TBM-related endocranial alterations other than APDIs (Fig 4B and Tables 5 and S4) and/or possible TB-associated non-endocranial bony changes (Fig 4B and Tables 5 and S6) were concomitantly recorded. Since the disease registered as the cause of death on the morgue record and/or death certificate may not have been the only medical condition present in the individuals surveyed in the Terry Collection, individuals identified to have died of causes other than TB could still have suffered from TB at death [5,6]. Thus, in the aforementioned 28 individuals with APDIs in the NTB group, the tuberculous origin of the recorded endocranial and non-endocranial lesions cannot be excluded. However, in the other 34 individuals with APDIs in the NTB group, where no signs of probable TB-related endocranial or non-endocranial bony changes were detected, the NTB origin (e.g., CNS infections other than TBM, brain tumors, and hemorrhages) of the noted lesions is much more likely.

The generated sensitivity and specificity estimate values also support that there is a positive association between APDIs and TB as the sensitivity of APDIs (i.e., 0.6581) is more than one minus the specificity of APDIs (i.e., 0.3212) (Table 4) [89]. It also means that the probability of presenting APDIs is higher in individuals with TB than in those without TB. However, it does not suggest that the majority of individuals with APDIs actually have TB. On the one hand, the calculated sensitivity estimate value indicates that if APDIs are present on the inner skull surface, there is a 65.81% probability of a true TB diagnosis. It implies that APDIs are suitable for screening purposes, since with their application as diagnostic criteria, quite a large number of true positives can correctly be identified. (It should be mentioned that APDIs could not be used to identify all TB cases, as APDIs can develop only in case of meningeal involvement that does not occur in all TB patients.) On the other hand, the calculated specificity estimate value suggests that if APDIs are not present, there is a 67.88% probability of the individual not having TB. Therefore, APDIs are not specific to TB, and consequently, they are not sufficient on their own to make a definitive diagnosis of TB due to the moderately high false positive rate (i.e., 0.3212). (It is not surprising if we consider that, besides TBM, numerous other medical conditions can also result in the development of APDIs [48,49]).

It should be noted that our research project has two major methodological limitations. 1) As it was already mentioned above, even if the morgue record and/or death certificate stated that an individual from the Terry Collection died of a NTB cause–and therefore, was enrolled into the NTB group–, it cannot be taken for granted that a concurrent active TB disease was not present, especially if we consider that most individuals from the Terry Collection lived in a time period and geographic region, in which TB was highly prevalent [22,50]. Furthermore, even if an individual from the Terry Collection died of TB, it cannot be excluded that there

was a coinciding NTB medical condition that could also result in the formation of APDIs on the inner skull surface. Unfortunately, the aforementioned sample selection problems–both for the TB group and NTB group–cannot be completely eliminated without knowing the full medical history of individuals that is barely available in case of identified skeletal collections from the pre-antibiotic era. Since these sample selection problems can affect the findings of statistical analyses (e.g., sensitivity and specificity estimates), in the future, it would be beneficial if we could extend and further improve our investigations by evaluating skeletons from a documented skeletal collection, in which the full medical history of individuals is known. 2) During the assessment of the 427 selected skeletons from the Terry Collection, only macro-morphological methods could be applied. Therefore, the classification of the examined skulls regarding the prominence of APDIs was based on comparison with three reference cases that, in our opinion, represented three different prominence stages of APDIs (i.e., very slight, slight, and pronounced)–the differentiation between the stages relied only on the location of the observed APDIs on the endocranial surface (Table 1). In the future, further investigations (e.g., clinical CT and micro-CT) are planned to reduce subjectivity and improve experimental reproducibility. These studies would apply methods that allow us to assign metric values to the different prominence stages (e.g., what percentage of a certain area of the inner skull surface should be covered by APDIs, how many APDIs should be present in a certain area of the inner skull surface, and what should be the average depth of APDIs present in a certain area of the inner skull surface). This would provide a more precisely described difference between them, and make the evaluation process more objective and experimentally reproducible.

Nonetheless, the main aim of our study was not to differentiate between the prominence stages but to strengthen the diagnostic value of APDIs in the identification of TBM in ancient human bone remains. Based on our results, APDIs can be used as diagnostic criteria for TBM in the paleopathological practice, but only when they are concurrent with other endocranial and/or non-endocranial bony changes likely associated with TB. The prudent utilization of APDIs provides paleopathologists with a stronger basis for diagnosing TBM in past human populations. Besides identifying individual TBM cases (based on the presence and association of skeletal lesions probably related to the disease); and thus, providing evidence for the existence of TBM in osteoarchaeological series, estimating how common the disease could be, and consequently, how substantial the impact it could have on past human populations is also an important aim of paleopathological studies [22,89]. Nevertheless, in most cases, it is very difficult to establish a definitive diagnosis of TB; and thus, to specify exactly which individuals suffered from the disease in an osteoarchaeological series, based only on the observable bony changes, since the majority of the endocranial and non-endocranial skeletal lesions that can be considered as diagnostic criteria for TB are not specific to the disease [22,89]. It must always be remembered that in the majority of cases, the course of TB is not long enough to allow bony changes to develop in the individual. Even if the skeletal system becomes involved, the manifestation of the disease can vary individual by individual and/or over time [22,89]. Therefore, if we consider only individual cases, more restrictively only those displaying skeletal changes that are thought to be pathognomonic for TB, when assessing the prevalence of the disease in past human populations, we will invariably underestimate it [22,89]. However, the simultaneous assessment of sensitivity and specificity estimate values for a wide range of bony changes likely associated with TB (not only the pathognomonic ones) provides us with a more sensitive means of assessing TB frequency, and consequently, of quantitatively estimating the impact of the disease on past human populations [22,89]. In the future, further investigations regarding the sensitivity and specificity estimates of the TBM-related endocranial alteration types (not only APDIs but GIs, ABVIs, and PAs) are planned to further highlight their diagnostic value in the paleopathological practice.

Similar to the results of our two previously published studies that were also performed on skeletons from the Terry Collection [23,24], the findings of our current paper may draw physicians' attention to the rather high prevalence of meningeal involvement in TB patients. Although the vast majority of individuals from the TB group were recorded to have died of pulmonary TB (S1 Table), more than two-thirds of them exhibited at least one out of the four examined endocranial alteration types probably related to TBM–about one-third of them displayed GIs that can be considered as specific signs of the disease [23]. These cases, along with the results of some autopsy studies [69], indicate that involvement of the CNS could be quite common in pulmonary TB patients even if they do not present with neurological signs and symptoms. If physicians become more cautious about the above, hopefully, they will check more and more pulmonary TB patients for meningeal involvement. This way the establishment of an early, accurate diagnosis and the initiation of a prompt, adequate treatment can be achieved [66,74]–these can be crucial in determining the clinical outcome of TBM, one of the most devastating clinical manifestations of TB with high short-term mortality and substantial excess morbidity among survivors [66,74].

## Supporting information

**S1 Table. Basic biographic data of individuals in the TB group (N = 234).** (MR = morgue record; DC1 = death certificate primary; DC2 = death certificate secondary; DC3 = death certificate tertiary; c. = circa; F = female; M = male; TB = tuberculosis; APDIs = abnormally pronounced digital impressions; 0 = not exhibiting APDIs; 1 = exhibiting very slight stage APDIs; 2 = exhibiting slight stage APDIs; 3 = exhibiting pronounced stage APDIs).
(PDF)

**S2 Table. Basic biographic data of individuals in the NTB group (N = 193).** (MR = morgue record; DC1 = death certificate primary; DC2 = death certificate secondary; DC3 = death certificate tertiary; c. = circa; F = female; M = male; TB = tuberculosis; NTB = non-tuberculous; APDIs = abnormally pronounced digital impressions; 0 = not exhibiting APDIs; 1 = exhibiting very slight stage APDIs; 2 = exhibiting slight stage APDIs; 3 = exhibiting pronounced stage APDIs).
(PDF)

**S3 Table. Individual data of cases exhibiting APDIs regarding other probable TBM-related endocranial alterations in the TB group (Σ = 154).** (TB = tuberculosis; TBM = tuberculous meningitis; APDIs = abnormally pronounced digital impressions; ABVIs = abnormal blood vessel impressions; PAs = periosteal appositions; GIs = granular impressions; + = present;– = not present).
(PDF)

**S4 Table. Individual data of cases exhibiting APDIs regarding other probable TBM-related endocranial alterations in the NTB group (Σ = 62).** (TB = tuberculosis; NTB = non-tuberculous; TBM = tuberculous meningitis; APDIs = abnormally pronounced digital impressions; ABVIs = abnormal blood vessel impressions; PAs = periosteal appositions; GIs = granular impressions; + = present;– = not present).
(PDF)

**S5 Table. Individual data of cases exhibiting APDIs on the inner skull surface regarding probable TB-related non-endocranial lesions in the TB group (Σ = 154).**
(TB = tuberculosis; APDIs = abnormally pronounced digital impressions; PNBFs = periosteal new bone formations; HPO = hypertrophic pulmonary osteopathy; + = present;– = not

present).
(PDF)

**S6 Table. Individual data of cases exhibiting APDIs on the inner skull surface regarding probable TB-related non-endocranial lesions in the NTB group ($\Sigma$ = 62).**
(TB = tuberculosis; NTB = non-tuberculous; APDIs = abnormally pronounced digital impressions; PNBFs = periosteal new bone formations; HPO = hypertrophic pulmonary osteopathy; + = present;– = not present).
(PDF)

## Author Contributions

**Conceptualization:** Olga Spekker.

**Data curation:** Olga Spekker, David R. Hunt.

**Formal analysis:** Olga Spekker, László Paja.

**Funding acquisition:** Olga Spekker, György Pálfi.

**Investigation:** Olga Spekker.

**Methodology:** Olga Spekker.

**Project administration:** Olga Spekker.

**Resources:** David R. Hunt.

**Supervision:** David R. Hunt, Erika Molnár, György Pálfi, Michael Schultz.

**Visualization:** Olga Spekker, William Berthon.

**Writing – original draft:** Olga Spekker, David R. Hunt, Michael Schultz.

**Writing – review & editing:** Olga Spekker, David R. Hunt.

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
