## [Decision Letter · Decision Letter 0]

25 Jan 2021

PONE-D-20-34541

Tracking down the White Plague. Chapter three: Revision of endocranial abnormally pronounced digital impressions as paleopathological diagnostic criteria for tuberculous meningitis

PLOS ONE

Dear Dr. Spekker,

Thank you for submitting your manuscript to PLOS ONE. After careful consideration, we feel that it has merit but does not fully meet PLOS ONE’s publication criteria as it currently stands. Therefore, we invite you to submit a revised version of the manuscript that addresses the points raised during the review process.

Please adjust the manuscript according to the suggestions for improvement brought forward by the reviewers. In case this is not possible, please discuss the reasons.

We look forward to receiving your revised manuscript.

Kind regards,

Michael C Burger, M.D.

Academic Editor

PLOS ONE

Journal Requirements:

2.Thank you for submitting the above manuscript to PLOS ONE. During our internal evaluation of the manuscript, we found significant text overlap between your submission and the following previously published works, some of which you are an author.

- https://journals.plos.org/plosone/article?id=10.1371%2Fjournal.pone.0230418 ("Tracking down the White Plague: The skeletal evidence of tuberculous meningitis in the Robert J. Terry Anatomical Skeletal Collection" by Spekker et al., 2020)

- https://journals.plos.org/plosone/article?id=10.1371/journal.pone.0238444 ("Tracking down the White Plague. Chapter two: The role of endocranial abnormal blood vessel impressions and periosteal appositions in the paleopathological diagnosis of tuberculous meningitis" by Spekker et al., 2020)

Please revise the manuscript to rephrase the duplicated text, cite your sources, and provide details as to how the current manuscript advances on previous work. Please note that further consideration is dependent on the submission of a manuscript that addresses these concerns about the overlap in text with published work.

Reviewers' comments:

Reviewer's Responses to Questions

**Comments to the Author**

1. Is the manuscript technically sound, and do the data support the conclusions?

Reviewer #1: Partly

Reviewer #2: Yes

2. Has the statistical analysis been performed appropriately and rigorously? 

Reviewer #1: Yes

Reviewer #2: Yes

3. Have the authors made all data underlying the findings in their manuscript fully available?

Reviewer #1: Yes

Reviewer #2: Yes

4. Is the manuscript presented in an intelligible fashion and written in standard English?

Reviewer #1: Yes

Reviewer #2: Yes

5. Review Comments to the Author

Reviewer #1: Comments to authors

This paper studies the presence of abnormally pronounced digital impressions (APDIs), which are alterations to the endocranial surface, and tuberculosis (TB) diagnosis in skeletons from the Terry collection - a known age and sex skeletal collection. Here the frequencies of the bone changes are compared in a group of cases that had cause of death listed as TB and a control group that had nonTB cause of death.

It is of great importance when trying to understand the evolution and history of TB that we are able to estimate the presence and impact of the disease in past populations. When in the distant past only skeletons are available as evidence of its presence and therefore we need to have well-developed and tested paleopathological methods as diagnostic tools to ensure precise estimates of disease frequencies in paleoepidemiological studies. This paper studies the diagnostic value of one of many assumed TB bony indicators and therefore provides important insights that can be used in future research into TB in the past. However, some issues need to be considered and mentioned in the paper before I find it publishable.

Issues that need to be addressed

Page 3, Line 57-62: In the Introduction when mentioning studies to establish a more reliable and accurate paleopathological diagnosis of TB I think the paper by Dangvard Pedersen et al. in IJPP December 2019 https://doi.org/10.1016/j.ijpp.2019.01.001 is overlooked. This paper also as the current paper manuscript for review performs a case-control study using Terry collection skeletons as cases. In this study the frequencies of more types of potential TB bony indicators are studied and compared.

Selection of controls for the study

As with many of the studies of tuberculosis related bony changes using reference collections this study also uses skeletons from the same collection as controls. This is in my opinion wrong. The nonTB diagnosis in these skeletons is very unsure which is a problem that the authors draw way too little attention to as it is only discussed very little on page 16, lines 390-396. The skeletons are of people living in a time period first half of 20th century where TB was common, those skeletons that were listed as having nonTB cause of death could very likely have been infected with TB and also have had active bone involving TB though not mentioned because they died of something else. This makes the control group in this study very weak. Treating the nonTB cause of death individuals as a control group also makes the results of the statistical analyses rather weak as it does not make sense to compare the case and control groups when likely only some of the ‘controls’ are nonTB individuals and the proportion of such is unknown.

Ideally the control group was selected from a collection of skeletons with a much more solid nonTB proof as done in the 2019 IJPP Dangvard Pedersen et al. publication where controls where found in the Bass collection, which are much more recent skeletons of people living in second half of 20th century where TB was almost eradicated in the US. As further data collection as part of the present publication is not feasible at least an elaborate discussion of these issues - more extensive than in the current version of the manuscript - are very important to include.

Lack of thorough descriptions of criteria for recording APDIs and other TB bony indicators

For other researchers to be able to perform comparative studies of the frequency of APDIs in the same or other skeletal collections it is important that the criteria are well described for both how to record the absence and presence of the bony indicators in the study.

In the first evaluation round absence and presence of APDIs were recorded, but how the assessment was done is not described in detail – for a positive score were there a certain number of DIs present, where the DIs at least a certain size or did they need to be situated on a certain part of the endocranial surface?

As I understand only the skulls with positive APDIs were evaluated in the second round and here they were given scores according to stage of progression. The stages are well described but only based upon one reference skull for each stage, which as also mentioned by the authors on page 17, line 403 is not optimal. For the publication maybe the stages could be described as represented by perhaps three or five positives for each stage thereby also describing the span of pathological changes for each stage.

As many paleopathologists have experienced the recording of bony lesions are rather subjective so to ensure more valid and adding som objectivity to the recordings it would have been a good idea to re-evaluate all skeletons in the second round because after studying the three cases with different stages some of those that were borderline positive might have been scored differently.

I would also have liked to find a more detailed description of the criteria used for recording the other TB related bone changes. Here only references to other studies are given.

Statistical analysis

When studying the diagnostic value of a symptom it is of great importance to evaluate how well the symptom detects a given condition. Such is done by estimating the probability measures sensitivity - the probability of having the symptom when you are indeed sick - and specificity - the probability of not having the symptom when you are not sick. In case-control studies such can be calculated (see Dangvard Pedersen, 2019) when you have the registrations of presence and absence of the symptom – in this case APDIs – and you know whether individuals were suffering or not suffering from TB. As the aim is to evaluate the diagnostic value of APDIs I would like to see a more direct presentation and discussion of these diagnostic probability measures in the paper. I think this would add to and make the argumentation for the importance of the lesion type as a TB bony indicator more convincing. This should however be done in a way where the weakness of the control group is taken into account.

Discussion of results

The discussion in my opinion gives too much attention to a comprehensive description of what APDIs are and how they are formed in TB meningitis, which to some extent is more suited in an introduction. Here, instead I would liked to have seen further elaboration on the results in the light of other previous studies and discussions of future steps that could be done evaluating ADPIs or other diagnostic criteria for TB and further how to use such to estimate the presence and impact of TB in past populations.

Tables and figures

I find the presentation of the results in the text a bit confusing as well as the design of figures 3 and 4. It would be nice to have a table presenting summary statistics for the frequency counts for overview and please consider redoing figures 3 and 4 differently perhaps as bar charts.

Reviewer #2: This manuscript addresses the association of abnormally pronounced digital impressions (APDIs) with tuberculosis to assess their diagnostic value for tuberculosis in past human populations. The paper was a pleasure to read. It is extremely well written and easy to follow. The supplemental tables provide extensive data on individuals in the Terry Collection that will enable others to conduct additional research. The study is well designed and the data provided support the conclusion that APDIs cannot be used in isolation to identify tuberculous meningitis but, when used together with other endocranial or skeletal lesions associated with TB, they will provide a stronger basis for the diagnosis of tuberculosis and assessing its frequency in past populations. The authors do not try to overinterpret their findings or stretch the conclusions in any way, which I find refreshing. The bibliography is extensive and thorough.

My comments for improvement are minimal. My main substantive comment is that the authors could present more statistical analysis to provide a better understanding of these lesions. The introduction suggests that digital impressions are more common in children and adolescents, which implies that they could, thus, be more common in young adults than in middle or older adults. Is there a pattern evident in the Terry collection? Does one need to control for age in assessing the diagnostic value of these lesions for tuberculosis (which, of course, disproportionately affects younger people)?

In the Ethics Statement, I recommend that the authors avoid using the term “specimen” in association with these human remains. Simply refer to them as individuals (i.e., Individual ID number on p. 7, line 167) or as human skeletal remains. On p. 9, line 203, for example, it would be better to say “All human skeletal remains evaluated…” and line 206 should be altered as well.

Minor editorial comments:

p. 4, line 85: Suggest changing to “describe clinical pictures and, thus, morphological features…”

p. 11, line 264: insert a comma after “whereas”

p. 12, line 291: Suggest changing to “In 142 of the 154 individuals…”

p. 16, line 379-380: Suggest changing to “This result indicates…”

p. 16, line 383: Suggest changing to “This observation further supports…”

p. 17, line 406: Suggest changing to “This research would apply methods…”

6. PLOS authors have the option to publish the peer review history of their article (what does this mean?). If published, this will include your full peer review and any attached files.

Reviewer #1: No

Reviewer #2: No

---

## [Author Response · Author response to Decision Letter 0]

22 Feb 2021

Dear Dr. Michael C. Burger,

I am very thankful for the reviewers’ insightful and constructive comments regarding our manuscript entitled “Tracking down the White Plague. Chapter three: Revision of endocranial abnormally pronounced digital impressions as paleopathological diagnostic criteria for tuberculous meningitis” that was submitted to PLOS ONE (manuscript ID: PONE-D-20-34541). I am sure that the reviewers helped us to improve the quality of our manuscript. The main text has been modified following the reviewers’ suggestions, and the revised version of our manuscript has been uploaded to the submission site of PLOS ONE.

Responses to the suggestions:

1) Reviewer 1 mentioned that in the “Introduction” part of our manuscript, a paper (i.e., Pedersen et al., 2019) has been overlooked. We greatly appreciate Reviewer 1’s comment, as the aforementioned article contains a lot of information relevant for our manuscript – it was included into the reference list, and in the main text, the references were re-numbered accordingly.

2) Reviewer 1 commented that, similar to many other studies that use reference collections to determine the diagnostic value of probable TB-related bony changes in the paleopathological identification of the disease, we selected skeletons from the same anatomical collection (i.e., Terry Collection) as controls. However, in Reviewer 1’s opinion, this is wrong, as “the nonTB diagnosis in these skeletons is very unsure, which is a problem”. We agree with Reviewer 1 that even if the recorded cause of death of individuals surveyed in the Terry Collection may not have been TB, individuals could still have suffered from the disease but their death was attributed to another medical condition. It should also be noted that there is always the possibility that an inaccurate cause of death was registered on the morgue record and/or death certificate of individuals from the Terry Collection. Based on the above, it cannot be excluded that in some cases in our control group (NTB group), the observed APDIs could be resulted from TB even if the recorded cause of death was not TB. As Reviewer 1 mentioned, we already highlighted this problem in the “Discussion and Conclusions” part of our manuscript; however, we discussed it only very little. We agree with Reviewer 1 that we cannot know the exact proportion of such individuals in our control group (i.e., NTB group), who could have TB. We also agree with Reviewer 1 that in the Terry Collection, “the skeletons are of people living in a time period first half of 20th century where TB was common”, meaning in an extreme scenario, the majority if not all of them could have been infected with TB. However, being infected with TB does not mean that these individuals actually developed active disease during their life, since in the majority of cases, TB infection remains latent (~90%). Furthermore, the disease primarily affects the lungs (~80%) without involvement of the bones. Even if extra-pulmonary sites are affected, only some forms of the disease can – directly (i.e., skeletal TB) or indirectly (e.g., TB pleurisy or TBM) – lead to the formation of bony changes. TBM – that, besides many other medical conditions, can result in the development of APDIs on the endocranial surface (but not in all cases) – is a very rare manifestation of the disease, occurring in less than 1% of the active TB cases today. Even if we suspect that in the pre-antibiotic era, the frequency of TBM could have been higher than of today, in our opinion, the proportion of individuals with TBM could not be that high in the NTB group that would have substantially weaken our statistical analysis – there could not be a statistically extremely significant difference between the TB group and NTB group regarding the frequency of APDIs, if the proportion of individuals with active TB disease, and consequently, with TBM would have been very similar or the same in the two groups. Nonetheless, we agree with Reviewer 1 that this problem should not be overlooked when selecting individuals for the control group. It is exactly because of this that when constructing our TB group and NTB group, not only the primary, but the secondary and tertiary cause of death data of the death certificate, as well as the morgue record data were considered – if at least one of these indicated TB as even a possible cause of death, the individual was enrolled into the TB group (S1 and S2 Tables). This way, as much as we could, we tried to minimize the chance of an individual with active TB disease would have been enrolled into the NTB group. It should also be mentioned that, since the disease registered as the cause of death on the morgue record and/or death certificate may not have been the only medical condition present in the individuals surveyed in the Terry Collection, individuals identified to have died of TB could still have suffered from a NTB medical condition at death that could result the development of bony changes similar to or identical with that of probably related to TB (e.g., signs of rib periostitis, vertebral hypervascularization, and ABVIs). Unfortunately, the aforementioned sample selection problems – both for the TB group and NTB group – cannot be eliminated without knowing the complete medical history of individuals, which is barely available in case of identified skeletal collections from the pre-antibiotic era. Therefore, based on the above, we agree with Reviewer 1 that our control group was not perfect but in our opinion, it should not be considered as wrong. Reviewer 1 mentioned that in the future, we could extend and further improve our investigations by selecting individuals for our control group from another anatomical collection that contains “much more recent skeletons of people living in second half of 20th century where TB was almost eradicated in the US”. We are very grateful for this suggestion as, in our opinion, by performing such investigations, we could further strengthen the results of our current manuscript, especially if we could find an identified skeletal collection, where the complete medical history of individuals is known; and thus, we could almost entirely eliminate the aforementioned sample selection problems (unfortunately, even if the complete medical history would be known, there would be a very little chance that an individual would have had an undiagnosed or misdiagnosed TB disease). We hope that in the future, we will have a chance to evaluate such skeletons. To execute Reviewer 1’s comment, we included a more extensive discussion of the sample selection problems into the “Discussion and Conclusions” part of the main text.

3) Reviewer 1 noted that for improving the experimental reproducibility of our research, a more thorough description of criteria for recording APDIs and other probable TB-related bony changes would be necessary. To execute Reviewer 1’s suggestion, a more detailed description of the criteria used was inserted into the main text as Table 1 (for APDIs) and Table 2 (for other probable TB-related bony changes). We agree with Reviewer 1 that in its present form, our classification system for the prominence stages of APDIs is quite subjective, since we compared our cases to three reference cases from the same anatomical collection (i.e., Terry Collection) that, in our opinion, represent three different prominence stages of APDIs (mainly based on the location of the observed APDIs on the endocranial surface). In connection with the above, Reviewer 1 noted that “the stages could be described as represented by perhaps three or five positives for each stage thereby also describing the span of pathological changes for each stage”. We agree with Reviewer 1 that it would be more ideal if, besides the location of the observed APDIs on the endocranial surface (Table 1), we could consider more criteria (e.g., number of APDIs in a certain area of the inner skull surface and/or average depth of the observed APDIs in a certain area of the inner skull surface) in differentiating between the prominence stages of APDIs. However, at the time of research, only macromorphological evaluation could be performed on the selected skeletons from the Terry Collection, and in our opinion, only the location of the observed APDIs could provide sufficient basis for differentiating between the prominence stages of APDIs. (As an observation, usually, the larger the affected endocranial surface area was, the deeper and more in number the observed APDIs were. However, further research is needed to adequately include these criteria into the classification system.) As Reviewer 1 mentioned, we already highlighted the above problem in the “Discussion and Conclusions” part of our original manuscript. Nevertheless, in the revised version, we tried to emphasize the importance of future investigations (e.g., clinical CT and micro-CT) that could help us to reduce subjectivity and improve experimental reproducibility. These investigations would apply methods that allow us to assign metric values to the different prominence stages (e.g., what percentage of a certain area of the inner skull surface should be covered by APDIs, how many APDIs should be present in a certain area of the inner skull surface, and what should be the average depth of APDIs present in a certain area of the inner skull surface); and thus, provide more precisely described difference between them, and make the evaluation process more objective. Although right now re-evaluation of the selected skeletons from the Terry Collection is not feasible, we agree with Reviewer 1 that in the future, following the determination of further classification criteria for the different prominence stages of APDIs, it would be a good idea to re-assess the skeletons. Even if our definitions and results regarding the different prominence stages of APDIs can only be considered as preliminary, Reviewer 1’s comments strengthened us that it is a topic that is worth to be studied in the future. Nonetheless, the main aim of our study was not to differentiate between the prominence stages of APDIs but to strengthen their diagnostic value in the identification of TBM in ancient human bone remains, which we did in our opinion.

4) Reviewer 1 advised that the diagnostic probability measures sensitivity and specificity should be calculated for APDIs, since “when studying the diagnostic value of a symptom it is of great importance to evaluate how well the symptom detects a given condition”. We greatly appreciate Reviewer 1’s comment as the calculated sensitivity and specificity estimate values further strengthened our results regarding the diagnostic value of APDIs in the paleopathological identification of TB; and, as Reviewer 1 already predicted, they made our argumentation more convincing. Following Reviewer 1’s suggestion, we supplemented our original manuscript with a more direct presentation (Table 4) and discussion of these diagnostic probability measures. 

5) Reviewer 1 noted that “the discussion in my opinion gives too much attention to a comprehensive description of what APDIs are and how they are formed in TB meningitis, which to some extent is more situated in an introduction”. In the “Introduction” part of our manuscript, we mentioned that the main aim of our study is to expend the knowledge and understanding on the development of APDIs, as well as to improve their paleopathological interpretation along with strengthening their diagnostic value in the identification of TBM in past human populations. In our opinion, to completely accomplish the aforementioned aim, a detailed, step by step description of how (i.e., through exactly which pathological processes) TBM can result in the formation of APDIs is vital, since it helps the reader to understand why these bony changes can be considered as diagnostic criteria for TBM in the paleopathological practice in the first place. In the paleopathological literature, there has been no such detailed description of the development of APDIs; and therefore, we think that it is crucial to provide a solid background in the formation of APDIs based on the medical literature from the pre-antibiotic era and today before discussing in detail our findings from the Terry Collection. Reviewer 2 highlighted that “the bibliography is extensive and thorough” that implies us that Reviewer 2 found it as one of the strengths of our manuscript. Since it was one of our aims to provide a detailed, step by step description of how TBM can contribute to the formation of APDIs, in our opinion, this description fits more in the “Discussion and Conclusions” part of our manuscript rather than in the “Introduction” part. It should be noted that the current manuscript is the third part of an article series (Spekker et al., 2020a – https://journals.plos.org/plosone/article?id=10.1371/journal.pone.0230418, Spekker et al., 2020b – https://journals.plos.org/plosone/article?id=10.1371/journal.pone.0238444). This article series presents results about the evaluation of the diagnostic value of four endocranial alteration types (i.e., GIs, ABVIs, PAs, and APDIs) probably related to TBM. When publishing our results from the Terry Collection regarding GIs (Spekker et al., 2020a), and ABVIs and PAs (Spekker et al., 2020b), we followed the same logic – i.e., to provide a solid medical background for our findings from the Terry Collection, we gave a very detailed description of how TBM can lead to the development of the aforementioned endocranial alterations. Based on the above, if Reviewer 1 agrees, we would not like to shorten and replace this part in our current manuscript about APDIs. Reviewer 1 also mentioned that “I would liked to have seen further elaboration on the results in the light of other previous studies and discussions of future steps that could be done evaluating APDIs or other diagnostic criteria for TB and further how to use such to estimate the presence and impact of TB in past populations”. Following Reviewer 1’s guidelines, we supplemented the “Discussion and Conclusions” part of our original manuscript.

6) Reviewer 1 commented that the presentation of the results and the design of Figures 3 and 4 are a bit confusing. Reviewer 1 suggested that “it would be nice to have a table presenting summary statistics for the frequency counts for overview”. Following Reviewer 1’s suggestion, three tables were created and inserted into the main text: Table 3 – summary of the statistical results, considering the different stages of the prominence of APDIs; Table 4 – summary of the statistical results, considering the diagnostic sensitivity and specificity of APDIs, and Table 5 – summary of the statistical results, considering the co-occurrence of APDIs with other likely TB-related lesion types. Reviewer 1 also mentioned that we should “consider redoing figures 3 and 4 differently perhaps as bar charts”. We agree with Reviewer 1 that the aforementioned figures could be confusing, but in our opinion, it was not because of the design of the figures but the figure legends that were not appropriately phrased. To execute Reviewer 1’s suggestion regarding Figures 3 and 4, we rephrased their figure legends – hopefully, the new descriptions will be more adequate and eliminate confusion. As for the design of Figures 3 and 4, if Reviewer 1 agrees, we would like to keep the original format (i.e., double pie chart). As we already mentioned above, the current manuscript is the third part of an article series about the evaluation of the diagnostic value of four endocranial alteration types (i.e., GIs, ABVIs, PAs, and APDIs) probably related to TBM. The diagnostic value of all the above-mentioned lesion types was assessed in the same anatomical collection (i.e., Terry Collection), considering the same aspects (i.e., frequency, macromorphological characteristics, co-occurrence with other probable TB-related lesion types). In the second article (Spekker et al., 2020b), we examined ABVIs and PAs, and we used the same diagram design (i.e., double pie chart) to present their co-occurrence with other probable TB-related lesions. In our opinion, for comparison purposes, it would be important to keep this design for presenting the co-occurrence of APDIs with other probable TB-related lesions in our current manuscript, since for the readers, it would be easier to make comparisons between the different lesion types (i.e., ABVIs, PAs, and APDIs) discussed in separate parts of the article series. We decided to use the double pie chart format because we thought that it is the most appropriate one for our result presentation purposes – we still think so, and hope that we could eliminate confusion by rephrasing the figure legends, and Reviewer 1 will be satisfied with the modifications we made.

7) Reviewer 2 noted that “the authors could present more statistical analysis to provide a better understanding” of APDIs. As it was mentioned above, Reviewer 1 also asked us to include more statistics, i.e. calculate the sensitivity and specificity estimate values, as by doing so, it “would add to and make the argumentation for the importance of the lesion type as a TB bony indicator more convincing”. To execute the reviewers’ suggestions, the aforementioned diagnostic probability measures were generated and the results (Table 4) were discussed in detail. Reviewer 2 also mentioned that it would be an interesting aspect to consider the age at death of the examined individuals in assessing the diagnostic value of APDIs for TB. We agree with Reviewer 2 that further analyses regarding this aspect should be performed. Some preliminary results were already published and available online in the PhD dissertation of the first author of our manuscript (Spekker, 2018 – http://doktori.bibl.u-szeged.hu/id/eprint/9714/). These findings support Reviewer 2’s idea that APDIs “could be more common in young adults than in middle or older adults”, since APDIs occurred with the highest frequency among individuals under the age of 30 years both in the TB group and NTB group. Now, our research group works on a manuscript, in which a lot of statistical findings will be presented to further highlight the diagnostic value of not only APDIs, but GIs, ABVIs, and PAs, as well. For instance, the frequency of the four probable TBM-related endocranial lesion types will be compared in different age groups and between the sexes in both the TB group and NTB group, and between the two groups. We would like to publish the aforementioned results in a separate paper; therefore, we did not include them into our current manuscript.

8) Reviewer 2 suggested that in the “Ethics Statement”, we should not use the term “specimen”. Following Reviewer 2’s guidelines, the text was altered throughout the “Ethics Statement”.

9) Regarding Reviewer 2’s editorial comments, the text was changed accordingly.

10) It was noted in the decision letter that a significant text overlap between our current manuscript and two previously published works, i.e., Spekker et al., 2020a (https://journals.plos.org/plosone/article?id=10.1371/journal.pone.0230418) and Spekker et al., 2020b (https://journals.plos.org/plosone/article?id=10.1371/journal.pone.0238444) has been found. As it was already mentioned in the cover letter and above, our current manuscript is the third part of an article series – that was also indicated in its title (i.e., “Chapter three”). This article series is about the evaluation of the diagnostic value of four endocranial alteration types (i.e., GIs, ABVIs, PAs, and APDIs) probably related to TBM – Spekker et al., 2020a is about GIs, Spekker et al., 2020b is about ABVIs and PAs, and the current manuscript is about APDIs. The diagnostic value of all the above-mentioned lesion types was assessed in the same anatomical collection (i.e., Terry Collection), considering the same aspects – the frequency, the macromorphological characteristics, and the co-occurrence of GIs, ABVIs, PAs, and APDIs with each other and with other probable TB-related lesion types were examined applying the same macromorphological methods. All the four evaluated endocranial alteration types are related to TBM, the diagnostic value of all of them has been questioned or has not been assessed in the paleopathological literature. Therefore, in the relevant sections of the “Introduction”, “Material and Methods”, and “Discussion and Conclusions” part of our current manuscript, we had to discuss about the same literature background regarding TBM, the same skeletal material (i.e. 427 selected skeletons from the Terry Collection), the same macromorphological methods, the same problems in considering GIs, ABVIs, PAs, and APDIs as diagnostic criteria for TBM, and the same future research plans regarding these endocranial alterations. Nevertheless, when writing our original draft, we tried to focus on the macromorphological characteristics and development of APDIs, and their relation to TBM in the “Introduction” and “Discussion and Conclusions” parts (in our two previously published articles, we concentrated on GIs, ABVIs, and PAs considering the same aspects) to avoid self-plagiarism as much as possible. In the “Material” part of our current manuscript, we tried to give a more detailed description of the Terry Collection that was only shortly introduced in our two other works, and in the “Methods” part, we tried to focus on the definition of the three different stages of APDIs (this was not discussed in our two previously published articles), and we did not describe in detail the other probable TB-related bony changes but only mentioned references (these were discussed in detail in our two other works) – again, to avoid self-plagiarism as much as we could. However, for instance, we could not change the “Ethics Statement” that itself is about one and a half pages long, since APDIs were examined on the same individuals from the Terry Collection as GIs, ABVIs, and PAs; therefore, we had to keep the 427 specimen numbers unchanged. Furthermore, since the three works are very closely related to each other, we thought that it would make it easier for the readers to compare the results about the four different endocranial alteration types published separately (findings about GIs were discussed in Spekker et al., 2020a, results about ABVIs and PAs were discussed in Spekker et al., 2020b, and findings about APDIs are discussed in our current manuscript), if we would use a “template text” when describing our aims and objectives, as well as our results and their discussion. However, we were always cautious to use the appropriate lesion name the work was about and to mention the results (i.e., absolute and percentage frequencies, as well as χ2, df, and P values) about the lesion the work was about. Let us give three examples of how we used the aforementioned “template text”. As for the ‘Introduction’ part, in Spekker et al., 2020a: “The objectives of our paper are: 1) To macroscopically evaluate the selected skeletons from the Terry Collection for the presence of GIs; 2) To compare the frequencies of GIs between individuals recorded to have died of TB versus those identified to have died of causes other than TB; 3) To macromorphologically characterize GIs regarding the localization, extent, and number of lesions on the affected cranial bone(s); and 4) To evaluate the diagnostic value of GIs”. In our current manuscript: “The objectives of our paper are: 1) To macroscopically evaluate the selected skeletons from the Terry Collection for the presence of APDIs; 2) To compare the frequencies of APDIs between individuals recorded to have died of TB versus those identified to have died of causes other than TB; 3) To macromorphologically characterize APDIs regarding the stage of the prominence of lesions on the affected cranial bone(s); and 4) To evaluate the diagnostic value of APDIs”. (The same “template text” was used in the “Introduction” part of Spekker et al., 2020b.) Considering the “Results” part, in Spekker et al., 2020b: From a total of 427 skeletons evaluated in the Terry Collection, 67 (15.69%) exhibited PAs on the inner skull surface: 47 (20.09%) of 234 individuals recorded to have died of TB (S1 Table) and 20 (10.36%) of 193 individuals identified to have died of causes other than TB (S2 Table)”. In our current manuscript: “From a total of 427 skeletons evaluated in the Terry Collection, 216 (50.59%) exhibited APDIs on the inner skull surface: 154 (65.81%) of 234 individuals recorded to have died of TB (S1 Table) and 62 (32.12%) of 193 individuals identified to have died of causes other than TB (S2 Table)”. (Other “template sentences” were also used in the “Results” part of our current manuscript that can seem to be word-for-word copies of sentences from our two other works, but they contain results regarding APDIs instead of GIs, ABVIs or PAs.) As for the “Discussion and Conclusions” part, in Spekker et al., 2020b: “Of the 427 selected skeletons with sectioned skulls from the Terry Collection, about one-sixth exhibited PAs on the endocranial surface and PAs were registered in both the TB group and NTB group”. In our current manuscript: “Of the 427 selected skeletons with sectioned skulls from the Terry Collection, about one-half exhibited APDIs on the endocranial surface and APDIs were registered in both the TB group and NTB group”. (Other “template sentences” were also used in the “Discussion and Conclusions” part of our current manuscript that can seem to be word-for-word copies of sentences from our two other works, but they contain information regarding APDIs instead of GIs, ABVIs or PAs.) In the revised version of our current manuscript, we tried to rephrase sentences in the “Introduction”, “Material and Methods”, “Results”, and “Discussion and Conclusions” part that in our opinion, were word-for-word copies from our two previously published manuscripts (i.e., Spekker et al., 2020a and Spekker et al., 2020b). However, we did not completely rephrase the “template text” but, to lessen the number of matches between sentences from our current and other works, used synonyms for some words. In our opinion, these sections of our text should not be considered as self-plagiarism, as we were always cautious to use the lesion name “APDIs” and to describe or discuss the results (i.e., absolute and percentage frequencies, as well as χ2, df, and P values) regarding APDIs in these sentences. We hope that after reading our above argumentation, using the “template text” will not be considered as self-plagiarism anymore, and we will not have to completely rephrase the relevant sections in the “Introduction”, “Results”, and “Discussion and Conclusions” parts of our current manuscript (we could already use the “template text” in Spekker et al., 2020b, and it was never considered as self-plagiarism). We really think that keeping the text this way would substantially help the readers in comparing the results regarding the four different endocranial alteration types probably related to TBM (i.e., GIs, ABVIs, PAs, and APDIs) that were published in our separate articles. 

In the revised version of our manuscript, we tried to execute all suggestions of the reviewers, and avoid self-plagiarism as much as possible. I hope this new version will be suitable for publication in PLOS ONE.

Thank you again for the reviewers’ insightful and constructive comments and your editorial work!

Sincerely yours,

Dr. Olga Spekker, PhD

---

## [Editor Report · Decision Letter 1]

10 Mar 2021

Tracking down the White Plague. Chapter three: Revision of endocranial abnormally pronounced digital impressions as paleopathological diagnostic criteria for tuberculous meningitis

PONE-D-20-34541R1

Dear Dr. Spekker,

We’re pleased to inform you that your manuscript has been judged scientifically suitable for publication and will be formally accepted for publication once it meets all outstanding technical requirements.

Kind regards,

Michael C Burger, M.D.

Academic Editor

PLOS ONE
---

## [Editor Report · Acceptance letter]

12 Mar 2021

PONE-D-20-34541R1 

Tracking down the White Plague. Chapter three: Revision of endocranial abnormally pronounced digital impressions as paleopathological diagnostic criteria for
tuberculous meningitis 

Dear Dr. Spekker:

I'm pleased to inform you that your manuscript has been deemed suitable for publication in PLOS ONE. Congratulations! Your manuscript is now with our production department. 

Kind regards, 

on behalf of

Dr. Michael C Burger 

Academic Editor

PLOS ONE